# Long-wave infrared photothermoelectric detectors with ultrahigh polarization sensitivity

Mingjin Dai[1,3], Chongwu Wang[1,3], Bo Qiang[1], Yuhao Jin[1], Ming Ye[1], Fakun Wang[1], Fangyuan Sun[1], Xuran Zhang[1], Yu Luo [1] ✉ & Qi Jie Wang [1,2] ✉

Filter-free miniaturized polarization-sensitive photodetectors have important applications in the next-generation on-chip polarimeters. However, their polarization sensitivity is thus far limited by the intrinsic low diattenuation and inefficient photon-to-electron conversion. Here, we implement experimentally a miniaturized detector based on one-dimensional tellurium nanoribbon, which can significantly improve the photothermoelectric responses by translating the polarization-sensitive absorption into a large temperature gradient together with the finite-size effect of a perfect plasmonic absorber. Our devices exhibit a zero-bias responsivity of 410 V/W and an ultrahigh polarization ratio ($2.5 \times 10^4$), as well as a peak polarization angle sensitivity of 7.10 V/W•degree, which is one order of magnitude higher than those reported in the literature. Full linear polarimetry detection is also achieved with the proposed device in a simple geometrical configuration. Polarization-coded communication and optical strain measurement are demonstrated showing the great potential of the proposed devices. Our work presents a feasible solution for miniaturized room-temperature infrared photodetectors with ultrahigh polarization sensitivity.

Infrared (IR) photodetectors are highly desirable in various fields such as environment monitoring, thermal imaging, molecular fingerprinting, and free-space communication[1–3]. Polarization-sensitive IR photodetectors are especially attractive owing to their widespread applications in chemical analysis, biomedical diagnosis, and face recognition[4–6]. Up to now, there have been two main approaches to detect the state of polarization of the incident light, namely free-space and on-chip methods. For the free-space method, it usually needs a polarization-insensitive detector combing with a series of optical components, such as polarizers and waveplates. As a result, such method using bulky and complicated optical systems needs large space for polarization state detection, which is not able to meet the demand of miniaturization and ultrahigh-density integration of the next generation on-chip

optoelectronic systems[7,8]. Therefore, as the other approach for detecting the state of polarization, the polarization-sensitive photodetectors have attracted broad interests recently owing to their filterless configurations. In general, the polarization-sensitive photodetectors rely on anisotropic absorption of either natural or artificial materials, or the artificial/natural heterostructures. In the past two decades, two-dimensional (2D) materials have shown attractive applications in mid-/long-wave IR photodetectors[9–11]. For example, two-dimensional (2D) van der Waals materials with low-symmetric crystal structures and optical anisotropy, such as black phosphorus (BP)[12,13], tellurium (Te)[5,14], and palladium selenide (PdSe₂)[15,16], have shown great potential in developing high-performance polarization-sensitive photodetectors, especially in the mid-/long-wave IR spectral ranges.

[1]School of Electrical and Electronic Engineering, Nanyang Technological University, Singapore 639798, Singapore. [2]Centre for Disruptive Photonic Technologies, School of Physical and Mathematical Sciences, Nanyang Technological University, Singapore 637371, Singapore. [3]These authors contributed equally: Mingjin Dai, Chongwu Wang. ✉e-mail: luoyu@ntu.edu.sg; qjwang@ntu.edu.sg

In general, for polarization-sensitive photodetectors, the crucial figure of merit (FoM) for polarization sensitivity is the polarization ratio (PR), which is calculated using PR = $V_{max}/V_{min}$ (or PR=$I_{max}/I_{min}$), where $V_{max}$ ($I_{max}$) and $V_{min}$ ($I_{min}$) represent the maximum and the minimum linear-polarization-dependent photovoltages (photocurrents), respectively. However, most of the polarization-sensitive photodetectors based on intrinsic anisotropy of 2D van der Waals materials exhibit a small PR (1 < PR < 20)[8,12]. To further increase the PR value of 2D materials-based photodetectors, several strategies have been proposed, including the development of built-in electric field enhanced PR by forming heterojunction/homojunction such as vertical BP homojunction (PR = 35 at $\lambda$ = 1700 nm)[13] and lateral BP homojunction (PR = 288 at $\lambda$ = 1450 nm)[12], BP/MoS$_2$ heterojunction (PR = 100 at $\lambda$ = 3.5 μm)[17], and plasmonic nanoantenna assisted photothermoelectric (PTE) responses with a configurable polarity transition such as Gr/Au nanoantenna (PR = 1 → ∞/−∞ → −1 at $\lambda$ = 4.0 μm)[8] and PdSe$_2$/Au metamaterials (PR = 1 → ∞/−∞ → −1 at $\lambda$ = 5.3 μm)[7]. Although PR is a commonly used FoM for polarization-sensitive photodetectors in the literature, it doesn't indicate the polarization sensitivity accurately and completely as it neglects the sensitivity of the photodetector. In other words, both the responsivity/detectivity and PR should be considered together when characterizing the polarization sensitivity of a typical polarization-sensitive photodetector, because the responsivity/detectivity describes the absolute photoresponse of photodetectors. Therefore, simultaneously achieving a high responsivity/detectivity and a large PR is critical and need to be considered when designing a detector with high polarization sensitivity. However, it is still elusive to realize such high-polarization-sensitive photodetectors, especially operating in the mid-/long-wave IR regions.

Here, to tackle such a challenge, we proposed a properly designed IR detector with ultrahigh polarization sensitivity comprising a perfect plasmonic absorber with finite-size effect and one-dimensional Tellurium (Te) nanoribbon with a large Seebeck coefficient. Leveraging on the giant anisotropic absorption and the localized absorption induced by the finite-size effect of the IR perfect plasmonic absorber, the polarized incident lights are effectively converted into polarization-resolved photovoltage responses in the Te nanoribbon through the PTE effect. As a result, our proposed device embraces an ultrahigh polarization sensitivity with a peak polarization angle sensitivity of 7.10 V/W•degree at a wavelength of 8.0 μm, arising from a high responsivity of 410 V/W together with an ultrahigh PR (2.5 × 10⁴) at room-temperature. In addition, thanks to the geometrical configuration of PTE responses in as-proposed devices, the full linear polarimetry detection including power intensity, polarization angle, and degree of linear polarization, is also achieved using a properly designed three-port device. Furthermore, as a proof-of-concept demonstration of the ultrahigh polarization-sensitive photodetectors, we utilize the developed polarization-sensitive photodetector in a polarization coded communication and an optical strain measurement system showing great potentials for applications. Our results are promising for the next-generation infrared detectors with high performance such as room-temperature operation, bandgap-independent, devisable operation wavelength, and high polarization sensitivity.

## Results and discussion
### Design of polarization-sensitive IR photothermoelectric detectors

To achieve high polarization-sensitive photoresponses, two main aspects including a high optical anisotropy and a high photon-to-electron conversion efficiency are considered in our proposed IR PTE detectors. As shown in Fig. 1a, the structure of our proposed mid-IR PTE detector is made up of a mid-IR perfect plasmonic absorber and a 1D tellurium nanoribbon (Te NR). The perfect plasmonic absorber is made from an array of rectangular gold (Au) microstructure, a

dielectric spacer (Al$_2$O$_3$), and an optical thick Au backplate. To investigate the optical properties of the perfect plasmonic absorber, full-wave electromagnetic simulations were performed. The structural parameters of the perfect plasmonic absorbers are obtained using global optimizations (see Supplementary Fig. 1 and Table 1). The optical absorption peak of the perfect plasmonic absorber can be well designed at a specific wavelength in the IR regime. Both the electric field and the absorption density distributions are simulated with a polarization angle of 0° and 90°, respectively. As shown in Fig. 1b, for incident light with the polarization angle of 0°, the electric field on the Au nanostructures is near zero (Fig. 1b). For the incident light with the polarization angle of 90°, the maximum normalized electric field can reach up to 20 showing a dipole resonant mode (Fig. 1b, the top right)[18,19]. The maximum absorption density can reach up to 10 and the main absorption of light is located at the Au microstructure (Fig. 1b, the bottom right). In addition, the polarization angle dependent absorption at 8.0 μm of the perfect plasmonic absorber is plotted in Fig. 1c. The absorption (Abs.) as a function of the polarization angle can be well fitted by a sine function as:

$$\text{Abs.} = \frac{\text{Abs.}_{90}}{2} + \frac{\text{Abs.}_{90}}{2} \times \sin(2\theta - 90) \qquad (1)$$

where, Abs.$_{90}$ denotes the peak absorption intensity for the polarization angle of 90°, and θ denotes the polarization angle. Here, the Abs. is about 0% and 98% for the incident light with the polarization angle of 0° and 90°, respectively. This indicates a large optical anisotropy of the designed perfect plasmonic absorber, which is crucial for achieving a high polarization-sensitive PTE response. Furthermore, we experimentally demonstrated the polarization angle resolved absorption of the designed perfect plasmonic absorber. Here, three typical perfect plasmonic absorber (R$_1$, R$_2$ and R$_3$) are designed and their polarization angle resolved absorption spectra are measured (Supplementary Fig. 2). The experimentally measured absorption spectra agree very well with the simulation results. Next, we focus on the photothermal heating effect of the perfect plasmonic absorber[20]. Figure 1d plots the simulated absorption (blue) and calculated temperature increase ΔT (red) as a function of the wavelength with the polarization angle of 90°. The maximum ΔT at the resonance wavelength of 8.0 μm is about 5.85 K corresponding to an absorption of 98%. The absorption spectrum and ΔT display the same trend. Moreover, the simulated temperature distribution (Supplementary Fig. 3a) is in line with the absorption density distribution (Fig. 1b, the bottom right). These results indicate a linear relation between them. Apparently, ΔT exhibits a polarization angle dependent and a linear relation to the laser power, which are verified by the simulation results (Supplementary Fig. 3b).

Apart from the large optical anisotropy used for supporting a large PR, the thermoelectric material with a large Seebeck effect, a low thermal conductivity, and a high carrier mobility, is also desirable to achieve a high responsivity. Benefiting from the high tolerance for selection of thermoelectric materials for the as-proposed PTE mechanism, we select Te nanoribbon as the active material owing to its advantages, such as ultralow thermal conductivity (2.16 W/m•K) due to its heavy atom mass[21], a high Seebeck coefficient (413 μV/K) boosted by the quantum confinement effect induced sharp shapes of the density of states at band edges[22,23], and a good electrical conductivity owing to its narrow-bandgap[24]. Here, the Te NRs are synthesized using a hydrothermal method[25,26] and characterized using Raman spectrum (Supplementary Fig. 4a, b). The electrical transport behavior indicates a p-type semiconductor with a hole mobility of 993 cm$^2$/V•s for Te NR based field effect transistor (Supplementary Fig. 4c, d). Finally, we consider the PTE response of a typical device with a perfect plasmonic absorber integrated with Te NR under a global IR illumination at room-temperature. With a consideration of the heat conductance, radiation, and convection, the temperature distribution of the device in a large

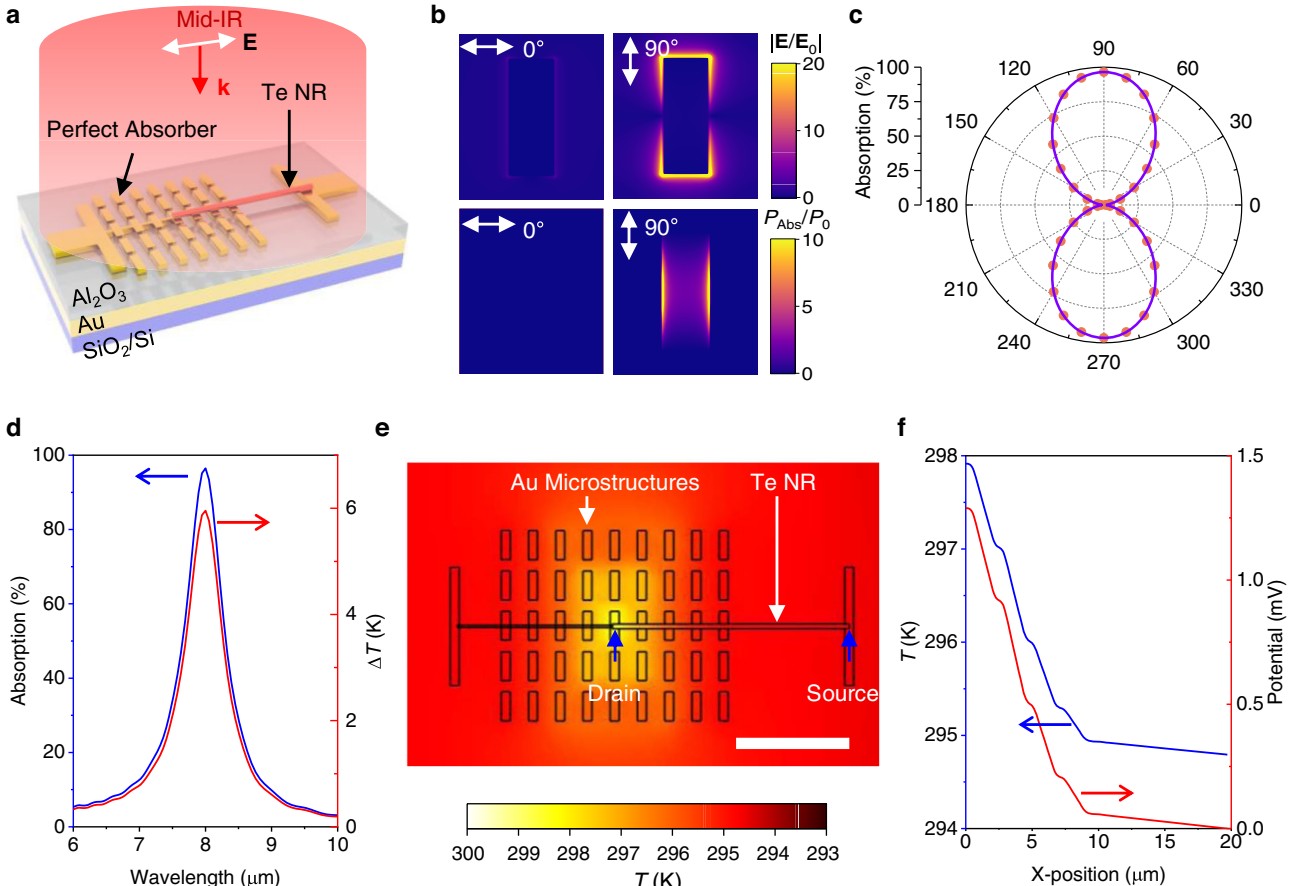

**Fig. 1 | Design of polarization-sensitive IR photothermoelectric detectors.**
**a** Device architecture of the resonance photothermoelectric detector. E and k indicate the electric field and vector of the incident electromagnetic wave. **b** Full wave simulations of electric field distributions (top) normalized to incident electric field, and power absorption density (bottom) is calculated by $P_{abs} = 1/2\omega\varepsilon'' |E|^2$, and is normalized by $P_0$, the incident power divided by the meta-molecule volume. **c** The linear polarization angle dependent absorption of the perfect plasmonic absorbers at the wavelength of 8 μm. Dots are simulated data and the line is fitting curve. **d** Simulated spectral absorption (blue) and temperature increase $\Delta T$ (red). The laser power is set to be 30 mW and the radius of laser spot is set to be 173 μm. **e** The thermal simulation of a typical device at peak absorption with input laser power of 30 mW, in the top view. Scale bar: 10 μm. **f** The corresponding temperature profile (blue) and potential profile (red) along the Te NR between drain and source electrodes.

---

scale was simulated as shown in Supplementary Fig. 5. Particularly, the temperature source is set according to the results of both photo-thermal simulations and experimental measurement. Figure 1e shows the temperature distribution surrounding the device indicating a large temperature gradient localized in the Au microstructure array. As shown in Fig. 1f, with an incident laser power of 30 mW, a large temperature gradient is built up (Fig. 1f, the blue line) along the Te NR direction, and the corresponding potential distribution is also plotted (Fig. 1f, the red line). The photovoltage response of about 1.30 mV is achieved, which is calculated by $V_{ph} = -S\Delta T$, where $S$ is the Seebeck coefficient and set to be 413 μV/K, and $V_{ph}$ is the photovoltage.

**Local absorption of the metamaterials with finite-size effect**
As shown in Supplementary Fig. 2, the experimentally measured polarization angle resolved absorption spectra show almost the same results as the full wave simulations, which are based on an infinite extended metamaterials obtained by periodically exploiting the elementary unit cell through the Floquet's theorem[27,28]. However, from the aspect of practical device fabrication, the finiteness of the array size and the boundary effects of the nano/micro-structure array should be considered[29]. Here, to investigate the finite-size effect of the perfect plasmonic absorber, we first prepared the perfect plasmonic absorbers with different array edge lengths (Fig. 2a) and measured their absorption spectra (Fig. 2b) under the polarization angle of 90°. The

absorption peaks for different array are all located at 8.0 μm, but the absorption peak intensity increases from 39% to 96% when the edge length of the Au microstructure array increases from 10 to 50 μm. This result arises from the symmetry broken at the array boundaries with a consideration of the dipole resonance of our plasmonic absorber[27]. On the other hand, the dipole resonance mode of each microstructure will vary with its location within the array, resulting in a nonuniform absorption density distribution[28]. Therefore, to further investigate this effect, we measured the absorption mapping under different polarization angles of a typical perfect plasmonic absorber array with a finite-size of $30 \times 30$ μm² (Fig. 2c). Figure 2d shows the absorption distribution of the perfect plasmonic absorber array with a finite size for mid-IR light at a resonance wavelength of 8.0 μm and with the polarization angle of 90°. As can be seen in Fig. 2d, for a finite-size array of the perfect plasmonic absorber, the maximum absorption is localized at the centra. The peak absorption intensity decreases from about 90% at the central position to 30% at the edge position, which is consistent with the previous analyses and results[28]. In addition, the same trend of the absorption distribution is observed under a polarization angle of 45° (Supplementary Fig. 6b), while the same trend was not observed for the polarization angle of 0° because the absorption intensity is near zero (Supplementary Fig. 6a). Although the finite size affects the total absorption of a perfect plasmonic absorber array, the localization of the absorption distribution can introduce a large

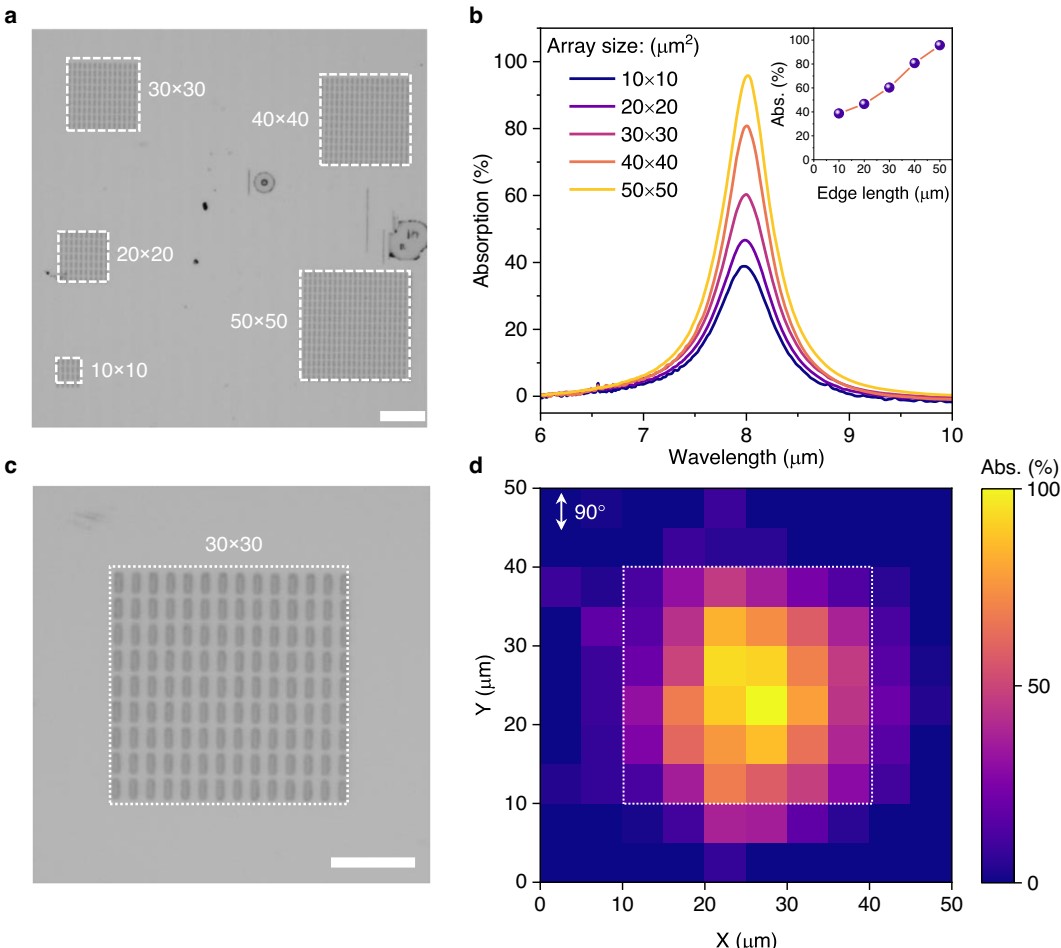

**Fig. 2 | Local absorption of the metamaterials with finite-size effect. a** Optical image of the metamaterials with different finite-sizes. Scale bar: 20 μm. **b** Measured absorption spectra of the metamaterials with different finite-sizes with a linear polarization angle of 90°. Inset shows the peak absorption density as a function of the array edge length. **c** Optical image of a perfect plasmonic absorber array with a finite size of 30 × 30 μm². Scale bar: 10 μm. **d** The corresponding absorption distribution of the metamaterials for infrared light with a wavelength of 8 μm and linear polarization angle of 90°. The dotted box indicates the boundary of the perfect plasmonic absorber array.

temperature gradient within the array, which can be utilized to further boost the PTE response in our device. Therefore, as illustrated in Fig. 1a and Fig. 1e, one electrode is set at the centra of the Au microstructure array, and the other electrode is outside of the array.

### Ultrahigh polarization sensitive photoresponses

As discussed above, the proposed PTE detector is properly designed to achieve an ultra-high polarization sensitivity by considering a high optical anisotropic ratio to support a high PR, a large temperature gradient, and a high Seebeck coefficient to support a high PTE responsivity, simultaneously. To experimentally demonstrate this, we fabricated a device as shown in Fig. 3a accordingly. In this device, the GND port indicates the ground terminal of one electrode, and ports 1 to 3 are used as the other electrodes with different channel lengths. Firstly, the $I_{ds}$-$V_{ds}$ curves for three ports show linear behaviors indicating good ohmic contacts between the Au electrode and Te NRs owing to their matched work functions (Supplementary Fig. 7a). Good ohmic contact in our device is also important for achieving a high responsivity because the barrier between electrode and thermoelectric material suppresses the PTE responses[30]. Then, under a global IR light illumination with a wavelength of 8.0 μm, the polarization resolved zero-bias photovoltage responses were measured for three ports (P₁, P₂, and P₃). As shown in Fig. 3b, all three ports show polarization resolved photovoltage ($V_{ph}$) responses,

which can be well fitted by:

$$V_{ph} = \frac{A}{2} + \frac{A}{2} \times \sin(2\theta - 90) \tag{2}$$

where, A denotes the maximum photovoltage at the polarization angle of 90°, and θ denotes the polarization angle. This indicates that the PR of three ports in our device is near infinite according to the fitting lines. In addition, the maximum photovoltage increases slightly with the increase of the channel length, verifying again that the dominating temperature gradient is localized within the Au microstructure array. To further verify this, simulation on the temperature distribution in such devices with different ports is carried out. As shown in Supplementary Fig. 8, for different ports with different channel lengths, the maximum temperature difference ($\Delta T$) changes slightly and the temperature falls sharply from the centra to the edge of the Au microstructure array due to the finite-size effect. Consequently, the responsivity decreases with the increase of the channel length (Supplementary Fig. 7b). The photovoltage responsivity changes from to 410 to 225 V/W, and the photocurrent responsivity calculated using the device resistance obtained from the $I_{ds}$-$V_{ds}$ curves in Supplementary Fig. 7a changes from 28.6 to 8.56 mA/W when port changes from P₁ to P₃. In contrast, the devices contacting only meta-atoms

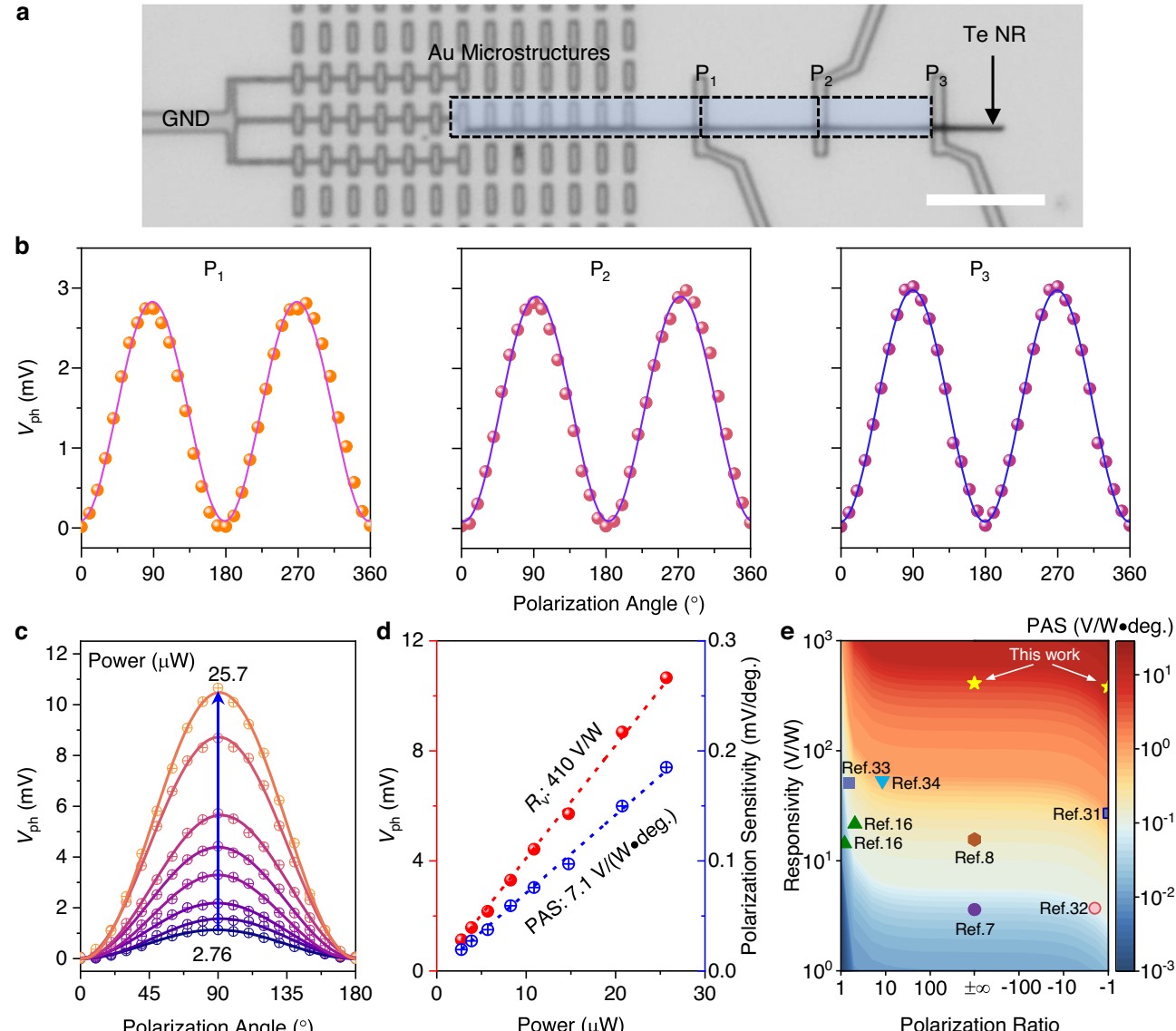

**Fig. 3 | Ultrahigh polarization sensitive photoresponses. a** Optical image of a photothermoelectric device with different channel lengths. The light blue area indicates the optical cross-section area. Scale bar: 10 μm. **b** Corresponding linear polarization angle dependent photoresponses obtained from three ports (P₁, P₂, and P₃). **c** The linear polarization angle dependent photoresponses under different light powers. Dots are measured data, and lines are fitting curves. **d** The light power dependent photovoltage and polarization sensitivity indicating a photovoltage

responsivity ($R_v$) of 410 V/W and a polarization angle sensitivity (PAS) of 7.1 V/ (W•degree). Dots are measured data and dashed lines are linearly fitting lines. **e** 2D contour map of the peak polarization angle sensitivity (PAS) for different polarization ratio and photovoltage responsivity. The PAS of our devices outperforms most of previously polarization-sensitivity photodetectors operating in the mid-/ long-wave infrared region.

in contact with the Te nanoribbon show a lower light absorption (Fig. 2b and Supplementary Fig. 10) and hence a lower photo-responsivity (8-20 V/W) (Supplementary Fig. 9) as compared to those of the devices with meta-atom arrays. This indicates that the meta-atoms even not in contact with Te nanoribbon play an essential role to enhance photoresponsivity through plasmonic-enhanced photon absorption and enlarged finite-size effect. Moreover, the polarization resolved photovoltage responses of P₁ were also measured with different incident light powers as shown in Fig. 3c. Here, the laser beam size is measured using our device with a short channel (20 μm) and the laser power density can be fitted well by using Gaussian distribution with two axis radii (1/e² intensity) as 173 and 173 μm (Supplementary Fig. 11). Considering the small size (channel length: 20–40 μm) of our devices, the peak intensity is used for calculating the light power

at device ($P_{Device}$) by[31]:

$$P_{Device} = \frac{2P_0 S}{\pi r_1 r_2} \qquad (3)$$

where, $P_0$ is the total power of the incident light, $r_1$ and $r_2$ are two axis radii of the Gaussian beam, and $S$ is the device optical cross-section area as shown in Fig. 3a. Here, the device optical cross-section area for ports 1–3 is 68, 102, and 136 μm², respectively, which is calculated by: $S = L \times P$, where $L$ is the channel length, $P$ is the period. When the power of the incident light at device P₁ increases from 2.76 to 25.6 μW, all the polarization resolved photovoltage responses can be fitted well using Eq. (2), and the maximum photovoltage A increases from 1.13 to 10.6 mV correspondingly. For different incident light powers, the device shows stable and repeatable photovoltage responses according

to the photo-switching test (Supplementary Fig. 12). Figure 3d plots the maximum photovoltages as a function of the incident light power showing a linear relation. By fitting the measured data, we can achieve a high responsivity of 410 V/W. Meanwhile, the transient photovoltage response shows a rise time of 176 μs and a decay time of 71 μs, indicating a −3dB bandwidth of 5.7 kHz of our device (Supplementary Fig. 13). Moreover, the detector also exhibits a low dark noise spectral density ($S_n$) down to 17 nV Hz$^{-1/2}$ at a high frequency range (over 5 kHz) corresponding to a noise-equivalent power of 0.04 nW Hz$^{-1/2}$ (Supplementary Fig. 14). The specific detectivity of the detector is calculated to be $1.7 \times 10^7$ Jones at room-temperature by $D^* = R\sqrt{S_{det}}/S_n$, where $S_{det}$ is the area of the detector, $R$ is the responsivity, and $S_n$ is the noise spectral density. Here, the detector area $S_{det}$ for Port 1 is 49 μm$^2$. Furthermore, the photoresponse of the device at low temperatures is also measured and shown in Supplementary Fig. 15. Coming from the efficient heat dissipation or high thermal conductivity at lower temperatures, the device exhibits a lower photovoltage response, which further certifies the PTE response mechanism of our proposed devices. Thanks to the configuration flexibility of the perfect plasmonic absorber, the proposed PTE detectors can also realize a bipolar response (PR = –1) with a well-designed device configuration (Supplementary Fig. 16). Comparing with the 2D Te nanosheet based device, the 1D Te NR based detector possesses a higher photovoltage response because of its lower heat capacity and smaller effective device area. With same device architectures, 2D Te nanosheet based device needs a higher incident light power to generate same photovoltage response as compared to the 1D Te NR based device. As a result, the 1D Te NR based device exhibits a higher responsivity (380 V/W) than that of 2D Te nanosheet based device (140 V/W) with the same incident power density. Furthermore, owing to its higher aspect ratio, the 1D Te NR based device can achieve a higher photoresponsivity and a smaller device footprint, simultaneously. This result indicates that 1D Te NRs are more advantageous than 2D Te nanosheets for PTE type detectors with the proposed device architecture in this work. On the other hand, other different 2D materials (PdSe$_2$, MoS$_2$, InSe) were used as the active materials for the proposed detectors. The responsivities are 27, 62, and 26 V/W for PdSe$_2$, MoS$_2$, and InSe based devices with the same device configurations, respectively (Supplementary Fig. 17). Our previous work also demonstrated that the 2D PdSe$_2$ nanosheet based detector shows a better performance than that of BP and Graphene based devices[7]. These results further indicate that the 1D Te NR is more suitable for the photothermoelectric detection with our proposed device architecture.

To further evaluate the polarization sensitivity of our proposed devices reasonably and accurately, a new FoM of the polarization angle sensitivity (PAS) is proposed by considering both the responsivity and the PR. As shown in Fig. 3c, the photovoltage is a function of polarization angle and can be expressed by Eq. (2). Therefore, the polarization sensitivity of photovoltage can be obtained by calculating the first-order derivative of photovoltage $V_{ph}$ versus polarization angle θ:

$$\frac{dV_{ph}}{d\theta} = A \times \cos(2\theta - 90) \qquad (4)$$

Therefore, the peak polarization sensitivity of photovoltage is localized at θ = 45°. The polarization sensitivity of photovoltage as a function of the incident light power is plotted in Fig. 3d and is fitted linearly, showing a peak PAS of 7.1 V/W•degree. More generally, both the responsivity ($R$) and the polarization ratio (PR) are given in most of previous reports. In this case, the A in Eq. (4) can be replaced by $R$ and PR. In addition, the $R$ is proportional to $V_{ph}$. As a result, the PAS can be calculated using:

$$PAS = R \times \frac{PR - 1}{PR} \times \cos(2\theta - 90) \qquad (5)$$

Based on this equation, the PAS as a function of both $R$ and PR is mapped and shown in Fig. 3e. As can be seen in Fig. 3e, the PAS increases when $R$ rises, and PR undergoes polarity transition from 1 to –1 with polarity transition. In addition, considering the polarization-dependent photoresponses described with Eq. (5), the PASs of the recently reported mid-/long-wave IR photodetectors are calculated and shown in Fig. 3e[7,8,31–34]. The maximal polarization ratio is about $2.5 \times 10^4$ when the incident light power at device is 25.7 μW, which is calculated based on the device voltage noise at a low frequency of 100 Hz (Supplementary Fig. 14). Comparing with other recent polarization-sensitive IR photodetectors, our proposed detectors exhibit a higher responsivity and a larger PR, as well as a higher PAS. In addition, the specific polarization angle detectivity (PAD) can also be calculated using the specific detectivity ($D^*$) of the detectors considering its proportional relation to $R$.

$$PAD = D^* \times \frac{PR - 1}{PR} \times \cos(2\theta - 90) \qquad (6)$$

As listed in Supplementary Table 2, the PADs are calculated for the mid-/long-wave polarization-sensitive photodetectors in both previous works and our work[7,8,16,31–34]. Our devices exhibit a high PAS of 7.1 V/W•degree (13.2 V/W•degree), which is one order of magnitude higher than those reported in previous studies, and a higher PAD of $2.9 \times 10^5$ Jones/degree ($3.8 \times 10^5$ Jones/degree) for the ultrahigh PR (PR = –1), indicating an ultrahigh polarization sensitivity. Comparing with commercially available IR detectors, the detectivity and the response speed of our detectors still need to be further improved (Supplementary Table 2).

## Full linear polarimetry detection

Although the polarization-sensitive photodetectors with a large PR have been achieved, based on either artificial nanostructures or natural semiconductors. Simultaneous detection of light intensity, polarization angle, and degree of linear polarization remains challenging. There are two main challenges for realizing the full linear polarimetry detection. The mechanism of most of existing polarization-sensitive photodetectors is based on the scalar anisotropic absorption, and hence their polarization angle resolved photoresponse signals overlap twice as the angle of linear polarized light changes from 0° to 180°[6,32]. Therefore, it is difficult to unambiguously detect the angle of linear polarized light in a single device. On the other hand, the detection of light intensity needs a polarization-insensitive photoresponse, which is more unrealistic for the previously reported detectors, although the polarization angle detection could be achieved by using the twist two-dimensional heterostructures or bulk photovoltaic effect in metasurface-mediated graphene detectors[6,31,35]. Here, leveraging on the configuration flexibility of the proposed perfect plasmonic absorber mediated photothermoelectric response, we design a suitable device with three ports for full linear polarimetry detection. As shown in Fig. 4a, three output ports ($P_0$, $P_1$, and $P_2$) are connected to the same ground terminal (GND) through three Te NRs and perfect plasmonic absorber arrays with different morphologies and orientations. In detail, a polarization-insensitive perfect plasmonic absorber is designed with an Au micro-disk array as illustrated in Supplementary Fig. 1 (right) and is used in port $P_0$ for the light intensity detection. The corresponding simulated electric field and absorption distribution at the designed resonance wavelength of 8.0 μm are shown in Supplementary Fig. 18a, b. More importantly, the experimentally measured polarization angle dependent absorption spectra show a polarization-insensitive absorption, which is the same as the simulated results (Supplementary Fig. 18c, d). In addition, two polarization-sensitive perfect plasmonic absorber arrays with a

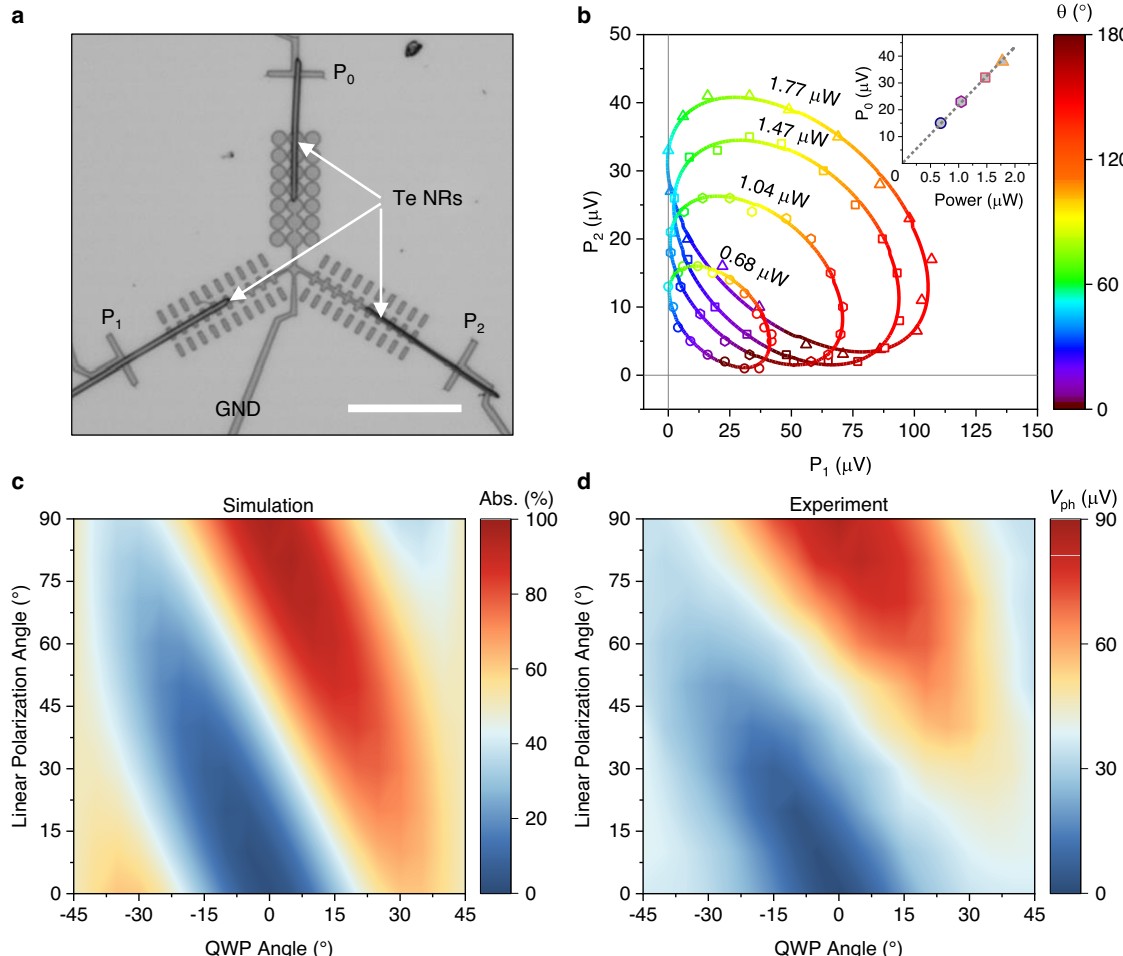

**Fig. 4 | Full linear polarimetry detection. a** Optical image of a typical device with three ports (P$_0$, P$_1$, and P$_2$) for full linear polarimetry detection. GND indicates the ground terminal. Scale bar: 10 μm. **b** Two-dimensional plots of P$_1$ and P$_2$ under different linear polarization angle θ and light powers. Inset shows the light power dependent photovoltage response of P$_0$. Dots are measured data, and lines are fitting curves. **c** Simulated polarization states dependent absorption of the meta-materials. **d** Experimental photovoltage responses under different polarization states. QWP quarter-wave plate.

relative orientation angle of 120° are used for ports P$_1$ and P$_2$ to extract the polarization angle θ. Figure 4b shows the two-dimensional plots of experimentally measured photovoltages of P$_1$ and P$_2$ under different incident light powers. The (P$_1$, P$_2$) pairs plotted with colored dots based on the polarization angle moves counterclockwise along a closed elliptical curve for a fixed incident light power. As the incident light power increases, the closed elliptical curve shifts towards the upper right, as shown in Fig. 4b. All (P$_1$, P$_2$) pairs are localized at the first quadrant owing to the unipolar responses of the designed devices. Apparently, there are some intersection points between the elliptical cures for different incident light powers. Therefore, it is necessary to distinguish these points with the port P$_0$, which is polarization insensitive and can help to extract the accurate polarization angle. Based on the polarization angle sensitive photovoltage as described by Eq. (2), the analytical expressions of the polarization angle θ dependent photovoltage response for three ports can be written as:

$$\begin{pmatrix} P_0 \\ P_1 \\ P_2 \end{pmatrix} = \begin{pmatrix} \frac{A_0}{2} \\ \frac{A_1}{2} \\ \frac{A_2}{2} \end{pmatrix} + \begin{pmatrix} \frac{A_0}{2} \\ \frac{A_1}{2} \times \sin(2(\theta + 30) - 90) \\ \frac{A_2}{2} \times \sin(2(\theta - 30) - 90) \end{pmatrix} \quad (7)$$

where A$_{0,1,2}$ refers to the maximum photovoltage outputs under a fixed incident power of port P$_0$, P$_1$, and P$_2$, respectively. The θ denotes the

polarization angle of the incident light referring to the horizontal axis as illustrated in Fig. 4a. Moreover, considering the linear relation between the incident light power and photovoltage output of port P$_0$ (inset in Fig. 4b), the incident light power intensity $P$ can be obtained from the output of port P$_0$ directly.

Beyond the light intensity, the polarization angle θ can be extracted using the measured signals at the ports P$_1$ and P$_2$ as:

$$\theta = \begin{cases} -\frac{1}{2}\tan^{-1}\left(-\frac{1}{\sqrt{3}}\frac{P_1' - P_2'}{P_1' + P_2'}\right) & \text{when } P_1' + P_2' < 0 \\ -\frac{1}{2}\tan^{-1}\left(-\frac{1}{\sqrt{3}}\frac{P_1' - P_2'}{P_1' + P_2'}\right) + 90 & \text{when } P_1' + P_2' > 0 \end{cases} \quad (8)$$

Here, both P$_1'$ and P$_2'$ are normalized photovoltage outputs using equation of $P_1' = \frac{2P_1}{A_1} - 1$ and $P_2' = \frac{2P_2}{A_2} - 1$, respectively. In addition, both A$_1$ and A$_2$ can be easily obtained by using the incident light power $P$ measured by port P$_0$ and corresponding responsivity (R$_1$ and R$_2$) for port P$_1$ and P$_2$. Now, the full linear polarimetry detection including both the intensity and polarization angle of the incident linear polarized light is realized by using our properly designed three-ports device.

However, the incident light is not always ideal linear polarized owing to its complex reflection, refraction, and scattering environments. Therefore, the degree of linear polarization (DoLP) is also another important figure of merit of the linear polarized light[36,37]. As discussed above, the port P$_0$ is polarization-insensitive, and hence the light intensity $P$ represents the total intensity of the incident light. For

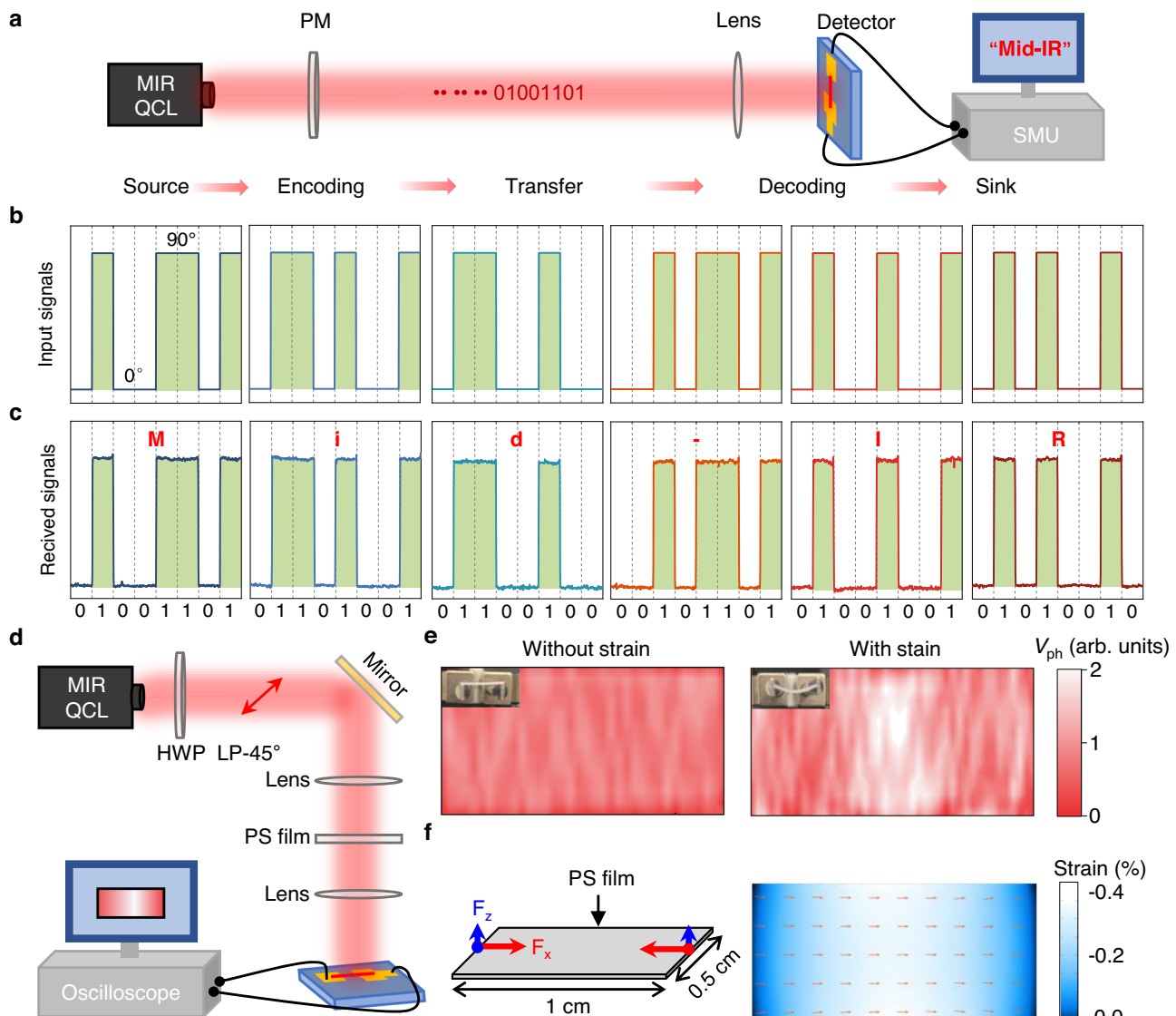

**Fig. 5 | Proof-of-concept applications in polarization coded communication and optical strain measurement. a** Schematic diagram of the experimental setup for polarization coded communication system. QCL quantum cascade laser, PM phase modulator, SMU source measurement unit. **b** Transited ASCII codes of "Mid-IR" letters encoded by the polarization angle (0° and 90°) of incident infrared light with a fixed power density. **c** Received signals by our polarization-sensitive detector. **d** Schematic diagram of the experimental setup for optical strain measurement

system. HWP half-wave plate. **e** Experimental measurements of photovoltage response for polarized infrared light transmitted a polystyrene (PS) film without and with strain. Insets show the optical images of the PS film without and with compression, respectively. **f** Simulation model (left) and simulated local strain distribution (right) of a PS film with a lateral size of $0.5 \times 1$ cm² and a thickness of 0.5 mm. $F_x$ and $F_z$ indicate the two forces loaded in x- and z-direction, respectively. The red arrows in strain distribution indicate the tangential strain directions.

example, the outputs of both $P_1$ and $P_2$ for a non-ideal linear polarized light must deviate from the closed elliptical curve of the ideal linear polarized light with a power intensity $P$. Despite this, the polarization angle θ can still be obtained by finding the intersection point between the closed elliptical curve and the straight line connecting the ($P_1$, $P_2$) and the center of the ellipse. Then, the DoLP can be obtained according to its relationship with the outputs of $P_1$ or $P_2$. To figure out the relation between DoLP of incident light and the photovoltage outputs of port $P_1$ or $P_2$, we use a quarter-wave plate to generate the non-ideal linear polarized light. Figure 4c shows the simulated absorption of the polarization-sensitive perfect plasmonic absorber as a function of linear polarization angle and quarter-wave plate (QWP) angle. Simultaneously, the photovoltage output of $P_1$ under a fixed light power intensity as a function of the linear polarization angle and the QWP angle are experimentally measured and shown in Fig. 4d. As can be seen, the experimentally measured $V_{ph}$ shows same

dependency on the linear polarization angle and the QWP angle as the absorption of the perfect plasmonic absorber. More importantly, the photovoltage responses of $P_1$ are monotonically increasing (decreasing) with the DoLP increases from 0 to 1 for θ = 90° (θ = 0°), indicating the ability to detect the DoLP of the incident light (Supplementary Fig. 19).

## Application demonstrations in polarization coded communication and optical strain measurement
Finally, to demonstrate the practical usage of our polarization-sensitive IR photodetectors with ultrahigh PR and ultrahigh polarization sensitivity, their applications in polarization coded communication and optical strain measurement are carried out. As shown in Fig. 5a, a mid-IR laser was used as the light source, and the input message was converted into American Standard Code for Information Interchange (ASCII) codes by changing the polarization angle (0° for

"0" and 90° for "1"). Subsequently, the message was transferred and encoded by the polarization-sensitive photodetector with an ultrahigh PR. At last, the signals output from the photodetector were transmitted to a terminal computer through a source-measure unit (SMU). Figure 5b shows the input ASCII signals of the letter "Mid-IR" encoding by the polarization angle. Figure 5c shows the received signals at our detector. The received signal exhibits perfect square waves and matches well with the original input information, indicating a high quality of reproduction of information transmission. Thanks to the ultrahigh PR of our proposed polarization-sensitive IR photodetectors, the optical communication with polarization-shift keying modulation scheme is realized, showing a great potential in applications of optical storage system, secure optical communication, and hyperspectral imaging[38-40]. In future works, the response speed of the proposed detector could be further improved, probably with improved design configurations, to realize a higher data rate.

In addition to the ultrahigh PR of our polarization-sensitive IR photodetectors, the ultrahigh polarization sensitivity supported by the high responsivity and large PR simultaneously could also offer great potential in optical strain measurement. Residual stress is an important factor of optical components that has a significant impact on their usage[41]. As a non-destructive measurement method to detect the strain in the optical components, the optical strain measurement based on the photoelasticity has widely been applied in the industry of glass manufacturing and photovoltaic panel manufacturing[42]. Here, taking the advantages of our polarization-sensitive IR photodetectors with ultrahigh polarization sensitivity, we apply it into the optical strain measurement system to evaluate the strain in a polystyrene (PS) film. As shown in Fig. 5d, the IR light from a laser source with a polarization angle of 45° transmitted through the PS film with two lenses and then was focused on the detector. The x-y position of the PS film without or with strain is controlled by two stepping motors and the photovoltage outputs from the detector are recorded using an oscilloscope simultaneously. Firstly, under a fixed light power and a polarization angle, the photovoltage distributions for both PS film with and without strain were measured and shown in Fig. 5e (left: without strain, right: with strain). Comparing with the $V_{ph}$ mapping for the PS film without strain, the $V_{ph}$ for compressed PS film shows a non-uniform distribution, and a higher $V_{ph}$ is localized at the center of the film. Then, to investigate the strain distribution in the compressed PS film, the simulation of the strain distribution is carried out using a finite element method. The model used for the simulation is shown on the left panel of Fig. 5f. According to the practical deformation as shown in Fig. 5f, two forces along x and z directions are loaded to the boundary of PS film with a lateral size of $0.5 \times 1\,cm^2$ and a thickness of 0.5 mm. As shown on the right panel of Fig. 5f, the simulated first principal strain shows a non-uniform distribution with a maximum negative strain of –0.4% at the center of the PS film. Comparing the $V_{ph}$ mapping shown on the right of Fig. 5e with the simulated local strain shown on the right of Fig. 5f, they show the same distribution profile, indicating the ability to test the local strain in the PS film of our polarization-sensitive IR photodetector. To further confirm that the $V_{ph}$ variation is caused by the local strain, a PS film with local positive strain induced by pre-embossing method is also used to measured (Supplementary Fig. 20a). As shown in Supplementary Fig. 20b, the measured $V_{ph}$ mapping shows a same distribution as the pre-embossed pattern of the "NTU" letters. In addition, the $V_{ph}$ is smaller for the positive strained area than that of the area without strain. This indicates that the positive and the negative strains in PS film will generates opposite change of the dielectric constant.

In conclusion, we proposed and demonstrated a properly designed polarization-sensitive IR detector with an ultrahigh polarization sensitivity. By means of large optical anisotropic absorption of the designed perfect plasmonic absorber, an ultrahigh polarization ratio ($PR = 2.5 \times 10^4$) is achieved in this work. Meanwhile, leveraging on the finite-size effect of the perfect plasmonic absorber array and the large Seebeck effect of the one-dimensional Te nanoribbon, a high responsivity is also realized. Taking advantages of the flexibility in device architecture design of the perfect plasmonic absorber, the proposed photothermoelectric detection mechanism in this work enables both unipolar ($PR = 2.5 \times 10^4$) and bipolar ($PR = -1$) polarization angle dependent photoresponses, as well as the full linear polarimetry detection including the power intensity, polarization angle, and degree of linear polarization of the incident light by using a single three-ports device. Notably, we also proposed two important FoM to evaluate the polarization sensitivity and the detectivity of a polarization-sensitive photodetectors: the polarization angle sensitivity with units of V/W•degree and polarization angle detectivity with units of Jones/degree. These two FoM will be more general than the existing polarization ratio, responsivity, and detectivity, especially for performance evaluation and comparison between different devices. Moreover, the proof-of-concept demonstration of applications in a polarization coded communication and an optical strain measurement system shows a great application potential of the as-proposed devices. Last but not the least, the detector performance in this work can be further improved, especially in terms of the response speed and the detectivity. Based on the photothermoelectric response mechanism, the performance of the detector can be further improved by systematically optimizing the electrode metal with lower contact barrier, the doping density of thermoelectric materials with higher Seebeck coefficients, the plasmonic metal microstructures with lower heat capacity and higher photothermal effect, the channel length matched better with the thermal decay length, and so on. Our investigations indicate that the integration of the artificial nanostructures with low-dimensional materials provides a promising routine for high-performance IR photodetectors with advanced functions.

## Methods
### Materials synthesis and characterization
For synthesis of the Te nanoribbons, the analytical-grade $Na_2TeO_3$ (0.0015 mol) and an amount of PVP were placed into distilled water (100 ml) under magnetic stirring to form a homogeneous solution. The resulting solution was poured into a Teflon-lined stainless-steel autoclave, together with an aqueous ammonia solution (25%) and hydrazine hydrate (80%). The autoclave was sealed and maintained at 180 °C for 3 h, and then cooled down to room temperature naturally. The resulting solid products were precipitated by centrifugation at 5000 r.p.m. for 5 min and washed three times with distilled water to remove residuals from the final product. The as-prepared Te nanoribbons are characterized by an optical microscope (Nikon) and a micro-Raman spectrometer system (WITec, Alpha300) equipped with a 532 nm laser source.

### Device fabrication and characterization
As the first step to device fabrication, a 200 nm thick gold thin film and a $Al_2O_3$ dielectric space layer with a typical thickness (200-270 nm) were firstly deposited onto a heavily p-doped silicon wafer grown with 285 nm thermal $SiO_2$ using e-beam evaporation. Then, electrodes and gold nanoantenna arrays were patterned on the chips using standard electron-beam lithography followed by thermal deposition of 5-nm-thick Cr and 50-nm-thick Au and lift-off process (submerging samples in acetone for 1 h). Thereafter, the Te nanoribbons were transferred onto the special position of the chip with electrodes and metamaterials by a dry-transfer method.

The optical absorption spectra were obtained using a Fourier Transform Infrared spectrometer (FTIR, Bruker) with a microscope (Thermo Fisher). Linear polarized lights are generated by using a linear polarizer. For the reflection spectra, the same sample without microstructures was used as the reference. The transmission is negligible due to the optically thick gold backplate. The absorbance spectra of

the metamaterials were calculated by using equation: Absorption = (1-Reflection) × 100%. The optical absorption distribution was obtained using the FTIR spectrometer integrated a microscope with an adjustable aperture to obtain a small light beam, and two stepping motors to control the sample position. The polarized photoresponse is measured by using a homemade photocurrent measurement system where the infrared light with different polarization statuses is obtained from a commerical quantum cascade laser (Daylight Solutions, MIRcat) with high linear polarization purity (>100:1) and tunable wavelength in the range of 7–11 μm combining series of half-wave plates, and then focused on the samples using a zinc selenide IR focusing lens with a focal length of 50 mm. The generated photovoltage was then recorded by a highly sensitive source-measure unit (Keysight, B2912A). For the low temperature photoresponse measurement, the device is mounted in a vacuum cryostat with a temperature controller. Here, we have selected three typical wavelengths in this work based on the operating wavelength of the half-wave plates and quarter-wave plate, 7.0 μm (Edmund, #85-121), 8.0 μm (Edmund, #85-122 and #85-115), and 9.0 μm (Edmund, #85-123). The voltage noise is measured by using a lock-in amplifier (Zurich Instruments, HF2LI).

## Simulation

The simulation of optical and photothermal properties of the metamaterials were done using Lumerical FDTD Solutions and HEAT packages. The simulated structure consists of a silicon substrate, SiO$_2$ (285 nm thickness), gold backplate (200 nm thickness), Al$_2$O$_3$ dielectric space layer (200–270 nm), gold antennas (50 nm thickness) and air. The power density absorption was calculated using the equation: $P_{abs} = 1/2\omega\varepsilon''|\mathbf{E}|^2$, where $\omega$ is the light frequency and $\varepsilon''$ is the imaginary part of the dielectric function. For the photothermal effect simulation in Heat package, an import heat source according to the optical absorption data obtained from FDTD simulation result is used as the heat input. For the simulation of temperature distribution of device with large scale (1 × 1 mm$^2$), COMSOL Multiphysics software with Heat Transfer Modules was used. The fixed temperature thermal boundary condition is applied at the surface of antennas according to the results from both the photothermal effect simulation in the HEAT package and experimental absorption distributions. For the simulation of strain distribution of the polystyrene film, COMSOL Multiphysics software with Solid Mechanics Modules was used. The density, Young modulus, and Poisson's ratio of the polystyrene film are set to be 930 kg/m$^3$, 0.9 × 10$^9$ Pa, and 0.38, respectively. Two forces (F$_x$: ±20 kN/m$^2$, F$_z$: 10 kN/m$^2$) were loaded at the two terminal boundaries and the free boundary condition was used for others.

## Data availability

Relevant data supporting the key findings of this study are available within the article and the Supplementary Information file. All raw data generated during the current study are available from the corresponding author Q.J.W. upon request. Source data are provided with this paper.

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

## Acknowledgements

This research was supported by National Research Foundation Singapore programme (NRF-CRP22-2019-0007 (Q.J.W.)), National Research Foundation Singapore Competitive Research Program (NRF-CRP22-2019-0006 (Y.L.)), A*STAR grant number A18A7b0058 (Q.J.W.), A20E5c0095 (Y.L., Q.J.W.) and A2090b0144 (Q.J.W.), and National Medical Research Council (NMRC) MOH-000927 (Q.J.W.). We thank Dr. C. Ge from Harbin Institute of Technology for his help in material synthesis and characterization.

## Author contributions

M.D. and C.W. contributed equally to this work. M.D. did the theoretical analysis and numerical simulation with assistance from B.Q. and M.Y. Both M.D. and C.W. did sample fabrication. M.D. carried out the device characterization with assistance from C.W., Y.J., F.W., X.Z., and F.S. All authors discussed the results. M.D., Y.L., and Q.J.W. wrote the manuscript with comments from all authors. Q.J.W. supervised the project.

## Competing interests

The authors declare no competing interests.
