## [Peer Review File · Nature Communications]

Long-wave infrared photothermoelectric detectors with ultrahigh polarization sensitivityREVIEWER COMMENTS

Reviewer #1 (Remarks to the Author):

In the current article, the authors demonstrated polarization-sensitive mid- or long-wavelength infrared (IR) detection based on the photothermoelectric effect of a tellurium nanoribbon. The polarization sensitivity stems from the aspect ratio of the ribbon and the anisotropic metamaterials. With many works already demonstrating polarization-sensitive or even full-Stokes IR detection, I don't think increasing the linear polarization ratio qualifies this article for Nature Communications. I have several concerns and questions for the authors.

(1). Ref. 7, published in Nature Communications early this year, shows that the same group demonstrated full-Stokes IR detection based on metamaterials and 2D materials. The 2D materials cover the entire metamaterial area, and each meta-atom contributes to the overall photothermoelectric responses. In sharp contrast, the current work has a Te nanoribbon only in contact with a few meta-atoms. In this sense, it is unclear whether the other meta-atoms could help overall IR detection. This ambiguity might lead to overcounted results in the overall sensitivity and responsivity. Even though the authors investigated the finite-size effect, experiments with metamaterials containing only meta-atoms in contact with the ribbon would be necessary and helpful.

(2) In Lines 270-273, the authors mentioned the responsivity of 2D Te nanosheets to be 380 v/w, while the 1D Te nanoribbon in this work is ~ 410 v/w. With such tiny differences, I am not sure that a 1D ribbon is more suitable, as claimed by the authors. Also, 2D nanosheets can be more compact with a smaller device footprint. They could keep the polarization sensitivity due to the anisotropy of the metamaterials.

(3) The demonstration of polarization-multiplexed communication is fine, but it is arguable about its applicability as the response time is a microsecond or a data rate of ~ 1 MHz.

Overall, the authors demonstrated wonderfully a polarization-sensitive long-wavelength IR detection. I will leave the decision of whether it deserves publication in Nature Communications to the editor.

Reviewer #2 (Remarks to the Author):

The authors demonstrated polarization sensitive IR detectors by integrating a plasmonic antenna metasurface absorber and Te nanoribbon with a large Seebeck coefficient. The device can be used to measure angle of linear polarization and degree of linear polarization, i.e. S_0 , S_1 , S_2 , three stoke parameters except S_3 , which is related to the degree of circular polarization. High performance polarization detectors and imaging sensors for IR wavelengths are of great interest for the research community. Yet, detection of linear polarization with very high selectivity is not a challenging topic anymore. Now in research field, major interests are focus on achieve the detection of circular polarization components, i.e., S_3 . Moreover, high performance detectors and imaging sensors in LWIR wavelength with low noise and high detectivity at room temperature are still badly needed.

I think the manuscript is not ready for publication in nature communication. I recommend major revision before considering it for publication, because the authors did not successfully prove the novelty and advantages of the proposed device configuration.

1) Polarization detector (polarimeter) and polarization detector arrays have been demonstrated based on different device configurations. The authors failed to mention in their introduction some compact solutions reported in literature, some of which in IR and LWIR wavelengths are already commercially available (<https://www.polarissensor.com/pyxis/>). This imaging sensor makes used of chip-integrated plasmonic nanogratings on the LWIR detector array to achieve high linear polarization selection to measure S_0 , S_1 , S_2 at each imaging pixel. In this manuscript, the authors make use of Plasmonic metasurface absorbers to introduce polarization dependent detector results, which is essentially the same concept. Detection of linear polarization with very high selectivity is not a challenging topic anymore. Now in research field, major interests are focus on achieve the detection of circular polarization components, i.e., S_3 . Many metasurface solutions have been provided to achieve complete measurements of full-stokes polarization detection. The following are just some examples, not all.

Pors, Anders, Michael G. Nielsen, and Sergey I. Bozhevolnyi. "Plasmonic metagratings for simultaneous determination of Stokes parameters." *Optica* 2.8 (2015): 716-723.

Li, W.; Coppens, Z. J.; Besteiro, L. V.; Wang, W.; Govorov, A. O.; Valentine, J. Circularly Polarized Light Detection with Hot Electrons in Chiral Plasmonic Metamaterials. *Nat. Commun.* 2015, 6, 8379

Bai, J.; Wang, C.; Chen, X.; Basiri, A.; Wang, C.; Yao, Y. Chip-Integrated Plasmonic Flat Optics for Mid-Infrared Full-Stokes Polarization Detection. *Photonics Res.* 2019, 7 (9), 1051– 1060,

Afshinmanesh, F.; White, J. S.; Cai, W.; Brongersma, M. L. Measurement of the Polarization State of Light Using an Integrated Plasmonic Polarimeter. *Nanophotonics* 2012, 1 (2), 125,

2) Materials with claimed large Seebeck coefficients have been studied extensively in literature, such as black phosphor. Some of the materials were reported with comparable or even higher values than the claimed value (>400 $\mu\text{V}/\text{K}$) for Te nanoribbons. The authors did not explain clearly and prove why they choose Te nanoribbons? What are the criteria for material selection and how the study proved the hypothesis?

3) The authors claimed a linear polarization extinction ratio of infinity, which is highly questionable. Usually, one obtains the linear polarization extinction ratio by taking the ratio between the detector output voltage/current for the favored linear polarization and that of its orthogonal polarization. A ratio of infinity means one reads exactly 0 for the orthogonal polarization in experiments, which does not seem to be reasonable if the measurement instrument has sufficient level of accuracy.

4) The D^* of the device is not very good compared with the commercially available LWIR detectors. The detection speed is not very impressive, either. I understand that this is just a prototype device. However, the authors did not provide analysis of the ultimate limits of device performance. Is it fundamentally possible to achieve better performance than the state of art LWIR detectors? If so, what does it take to get there?

5) When calculating of NEP and D^* , the authors estimated the power incident onto their devices using the ratio between the device area and the optical spot size. This method is a bit questionable because the device optical cross-section is usually larger than the device area. Besides, it is necessary for the authors to provide all the parameter values used for estimating NEP and D^* , including the device area A .

Reply to the Reviewers' Comments

We are grateful to the constructive comments and advice provided by the reviewers on our manuscript Nature Communications (ID: NCOMMS-22-43717. Title: Long-wave infrared photothermoelectric detectors with ultrahigh polarization sensitivity). In response to these comments, we have carried out further experiments and simulations, provided necessary discussions and analyses, and revised the manuscript and the supplementary information thoroughly. We believe that we have addressed all the comments raised by the reviewers.

The corresponding amendments addressing the referees' concerns have been marked with yellow background for easy reference in the revised manuscript (Revised Manuscript with marks) and revised Supplementary Information (Revised Supplementary Information with marks). Our replies to the comments by the reviewers and the corresponding changes are listed below.

The following are details of our point-by-point responses to reviewers' comments:

Reviewer #1:

Comments:

In the current article, the authors demonstrated polarization-sensitive mid- or long wavelength infrared (IR) detection based on the photothermoelectric effect of a tellurium nanoribbon. The polarization sensitivity stems from the aspect ratio of the ribbon and the anisotropic metamaterials. With many works already demonstrating polarization-sensitive or even full-Stokes IR detection, I don't think increasing the linear polarization ratio qualifies this article for Nature Communications. I have several concerns and questions for the authors.

Response to reviewer: We are grateful to the reviewer for the review and appreciate the reviewer's constructive suggestions/comments that have greatly helped us to improve the quality and depth of our work.

We agree with the reviewer that many research works have demonstrated polarization-sensitive or even full-Stokes IR detection. These efforts include the work done by Li et al. [*ACS Nano* 2020, 14, 12, 16634–16642] and by our group [*Nat. Commun.* 13, 4560 (2022)] for which we realized full-stoke detection. However, we would like to highlight that the detection of linear polarization with improved selectivity is still an important research topic,

as it has a number of important applications in polarization-coded optical communication, optical storage, and optical imaging and sensing. However, owing to the low polarization sensitivity ($< 1 \text{ V/W} \cdot \text{degree}$) of the reported detectors, their applications in the above areas are hindered significantly.

In this work, focusing on the linear polarization sensitivity, we not only realize an ultrahigh polarization ratio ($\text{PR} \rightarrow \infty$), but also a high photoresponsivity (410 V/W), thereby achieving a world-recorded ultrahigh polarization sensitivity ($7.1 \text{ V/W} \cdot \text{degree}$). The novelty and advantages of our work come from the following two aspects:

1: The configuration of our photodetectors is new. Previous works are mostly based on the intrinsic or the artificial nanostructure enhanced optical anisotropy of the low-dimensional crystal, whereas our work provides an efficient way to electrically readout the polarization state by achieving simultaneously the finite-size effect through an excellent plasmonic absorber and the large Seebeck effect from 1D Te nanoribbon. For the first time, the large temperature gradient induced by the finite-size effect is utilized to improve the thermoelectric response. On the other hand, the 1D Te nanoribbon with large Seebeck effect is used as the active materials for improving the thermoelectric response while reducing the device footprint down to the sub-wavelength scale (e.g. $3.4 \mu\text{m}$ for the wavelength of $8 \mu\text{m}$) in one direction, which shows great promise for compact devices.

2: The performance characterization of our photodetectors is new. Most of previous works just use polarization ratio to evaluate the performance of polarization-sensitive photodetectors which is not accurate enough, whereas our work proposed a combined method to evaluate simultaneously two important figures of merit (FoM) i.e. the polarization sensitivity and detectivity. As a result, our method is more accurate than all previous approaches, making the polarization-sensitive photodetector we designed highly suitable for a number of practical applications, e.g. 1) when applied to the polarization angle sensitivity (PAS), our photodetector can push the sensitivity limit to $7.1 \text{ V/W} \cdot \text{degree}$; 2) when applied to polarization angle detectivity (PAD) our photodetector has a sensitivity of $2.9 \times 10^5 \text{ Jones/degree}$. As we demonstrate experimentally, these two FoM (as proposed in our work) are more general than the polarization ratio (widely used in previous works), because the polarization ratio loses some important information such as responsivity and detectivity, especially for performance evaluation and comparison between devices with different materials and different configurations. With a consideration of both polarization ratio and responsivity, the proposed photodetectors in our work show an ultrahigh polarization sensitivity, which is about one order of magnitude higher than those reported in the literature.

Comment 1: Ref. 7, published in Nature Communications early this year, shows that the same group demonstrated full-Stokes IR detection based on metamaterials and 2D materials. The 2D materials cover the entire metamaterial area, and each meta-atom contributes to the overall photothermoelectric responses. In sharp contrast, the current work has a Te nanoribbon only in contact with a few meta-atoms. In this sense, it is unclear whether the other meta-atoms could help overall IR detection. This ambiguity might lead to overcounted results in the overall sensitivity and responsivity. Even though the authors investigated the finite-size effect, experiments with metamaterials containing only metaatoms in contact with the ribbon would be necessary and helpful.

Response to reviewer: We thank the reviewer for pointing out this important aspect which helps to improve the clarity of our manuscript. According to the reviewer's suggestion, we carried out additional experiments to measure the photoresponses of the device containing only meta-atoms in contact with 1D Te nanoribbon as shown in Supplementary Fig. 9. The simulated absorption power density is shown in Supplementary Fig. 10 in the revised Supplementary Information. Owing to the limited number of the meta-atoms, the absorption of the device becomes weaker leading to a lower photoresponsivity. Moreover, the decrease of the meta-atom number also reduces the thermal gradient induced by the finite-size effect, leading to further decrease of the photoresponsivity. This result indicates that the other meta-atoms even not in contact with the Te nanoribbon are essential to enhance the light absorption at the central region of the whole structure. Moreover, they also play an important role in increasing the contrast of thermal gradient through the finite-size effect. Both factors contribute to the increase of the photoresponsivity. To clearly show the finite-size effect of the metasurface array, we have added corresponding comparative discussions in the Revised Manuscript, also copied below:

On page 10 of revised Manuscript.

.....In contrast, the devices contacting only meta-atoms in contact with the Te nanoribbon show a lower light absorption (Fig. 2b and Supplementary Fig. 10) and hence a lower photoresponsivity (8-20 V/W) (Supplementary Fig. 9) as compared to those of the devices with meta-atom arrays. This indicates that the meta-atoms even not in contact with Te nanoribbon play an essential role to enhance photoresponsivity through plasmonic-enhanced photon absorption and enlarged finite-size effect.....

On page 10 of the revised Supplementary Information.

Supplementary Figure 9: Polarization-sensitive photoresponses of the devices with only contacting meta-atoms. a, b, Optical image (top) of the devices with only contacting meta-atoms and polarization-sensitive photoresponses (bottom). Scale bar: 30 μm . Dots are measured data, and red lines are fitting curves.

On page 11 of the revised Supplementary Information.

Supplementary Figure 10: Simulated absorption power density of the contacting meta-

atoms. Scale bar: 5 μm .

Comment 2: In Lines 270-273, the authors mentioned the responsivity of 2D Te nanosheets to be 380 V/W, while the 1D Te nanoribbon in this work is ~ 410 V/W. With such tiny differences, I am not sure that a 1D ribbon is more suitable, as claimed by the authors. Also, 2D nanosheets can be more compact with a smaller device footprint. They could keep the polarization sensitivity due to the anisotropy of the metamaterials.

Response to reviewer: We thank the reviewer for raising the concern on which we didn't describe clearly. In fact, the responsivities of 1D Te nanoribbon-based devices are 410 and 380 V/W for $\text{PR} \rightarrow \infty$ and $\text{PR} = -1$, respectively. On the contrary, the responsivity of 2D Te nanosheet based device with $\text{PR} = -1$ is only about 140 V/W, more than twice smaller than what we have achieved for 1D Te nanoribbon in this work. As shown in the Supplementary Fig. 16, the devices based on 1D Te nanoribbon and 2D Te nanosheet exhibit almost the same photovoltage response, but under different incident light powers. For a given photovoltage response, the incident power required for 2D nanosheet is almost linearly proportional to the width of the nanosheet. On one hand, a wider channel for the 2D Te nanosheet-based device is associated with a wider effective device area, and hence, a higher incident light power is required if the incident intensity is kept unchanged. On the other hand, a wider channel for 2D Te nanosheet-based device also leads to a higher heat capacity, and hence, a higher incident light power is required to make the temperature gradient unchanged within the channel. In contrast, the 1D Te nanoribbon based-device can realize simultaneously a higher photoresponse (i.e 410 V/W) and a smaller device footprint (i.e $3.4 \times 30 \mu\text{m}^2$), simultaneously. To clearly compare the performance between 1D Te nanoribbon and 2D Te nanosheet based devices, we have added corresponding comparative discussions in the Revised Manuscript, also copied below:

On pages 11-12 of the revised Manuscript.

.....Thanks to the configuration flexibility of the perfect plasmonic absorber, the proposed PTE detectors can also realize a bipolar response ($\text{PR} = -1$) with a well-designed device configuration (Supplementary Fig. 16). Comparing with the 2D Te nanosheet based device, the 1D Te NR based detector possesses a higher photovoltage response because of its lower heat capacity and smaller effective device area. With the same metasurface device architectures, the 2D Te nanosheet based device needs a higher incident light power to generate the same photovoltage response as compared to the 1D Te NR based device. As a result, the 1D Te NR based device exhibits a higher responsivity ($R = 380$ V/W) than that of 2D Te nanosheet based device ($R = 140$ V/W) with the same incident power density. Furthermore, owing to its higher aspect ratio, the 1D Te NR based device can achieve a higher

photoresponsivity and a smaller device footprint, simultaneously. This result indicates that 1D Te NRs are more advantageous than 2D Te nanosheets for PTE type detectors with proposed device architecture in this work.....

Comment 3: The demonstration of polarization-multiplexed communication is fine, but it is arguable about its applicability as the response time is a microsecond or a data rate of ~1 MHz.

Response to reviewer: We thank the reviewer for the constructive comments on the application demonstration example. The demonstration of polarization-coded communication is not only just to show the application potential of our polarization-sensitive photodetector, but also to illustrate the advantages of the near infinite polarization ratio in practical applications. We would like to highlight that, our photodetector with a near infinite polarization ratio is very useful for the polarization-shift keying modulation scheme that uses polarization states as an information-bearing unit to increase the overall spectral efficiency, as demonstrated in our manuscript. On the other hand, for polarization-coded communications in the mid-infrared, the main objective is not to achieve high-speed but more importantly to ensure security (i.e., achieving point-to-point free space laser communications) together with optical storage, imaging, and sensing potentials (i.e., hyperspectral imaging in the mid-infrared) in the night environment. To clarify on this point, we added corresponding discussions in the Revised Manuscript, also copied below:

On page 17 of the revised Manuscript.

.....Thanks to the near-infinite PR of our proposed polarization-sensitive IR photodetectors, the optical communication with polarization-shift keying modulation scheme is realized, showing a great potential in applications of optical storage system, secure optical communication, and hyperspectral imaging³⁸⁻⁴⁰. In future works, the response speed of the proposed detector could be further improved, probably with improved design configurations, to realize a higher data rate.

Comment 4: Overall, the authors demonstrated wonderfully a polarization-sensitive long-wavelength IR detection. I will leave the decision of whether it deserves publication in Nature Communications to the editor.

Response to reviewer: We are thankful to the reviewer for the constructive comments and appreciations of our work on realizing wonderful polarization-sensitive long-wavelength IR detectors.

Reviewer #2:

Comments:

The authors demonstrated polarization sensitive IR detectors by integrating a plasmonic antenna metasurface absorber and Te nanoribbon with a large Seebeck coefficient. The device can be used to measure angle of linear polarization and degree of linear polarization, i.e. S0, S1, S2, three stoke parameters except S3, which is related to the degree of circular polarization. High performance polarization detectors and imaging sensors for IR wavelengths are of great interest for the research community. Yet, detection of linear polarization with very high selectivity is not a challenging topic anymore. Now in research field, major interests are focus on achieve the detection of circular polarization components, i.e., S3. Moreover, high performance detectors and imaging sensors in LWIR wavelength with low noise and high detectivity at room temperature are still badly needed. I think the manuscript is not ready for publication in nature communication. I recommend major revision before considering it for publication, because the authors did not successfully prove the novelty and advantages of the proposed device configuration.

Response to reviewer: We are grateful to the reviewer for the review and appreciate the reviewer's constructive suggestions that helped us to improve the quality and depth of our work.

We fully agree with the reviewer that substantial efforts in research field have been focused on achieving the detection of circular polarization components, i.e., S3. These efforts include the work done by Wei et al. [*Nature Photonics* **17**, 171–178 (2023)] and by our group for which we realized full-stoke detection [*Nat Commun* **13**, 4560 (2022)]. However, the detection of the linear polarization with a very high selectivity is still a challenging topic, as the polarization sensitivity (meaning a high polarization ratio and a photoresponsivity or detectivity) in mid/long-wave infrared range in previous works is still relatively low, e.g. $<1 \text{ V/W}\cdot\text{degree}$.

In this work, focusing on linear polarization sensitivity, we not only realize an ultrahigh polarization ratio ($\text{PR} \rightarrow \infty$), but also a high photoresponsivity (410 V/W), thereby achieving a high polarization sensitivity (7.1 V/W•degree). The novelty and advantages of our work come from the following two aspects:

1: The configuration of our photodetectors is new. Previous works are mostly based on the intrinsic or the artificial nanostructure enhanced optical anisotropy of the low-dimensional crystal, whereas our work provides an efficient way to electrically readout the polarization state by achieving simultaneously the finite-size effect through an excellent plasmonic

absorber and the large Seebeck effect from the 1D Te nanoribbon. For the first time, the large temperature gradient induced by the finite-size effect is utilized to improve the thermoelectric response. On the other hand, the 1D Te nanoribbon with large Seebeck effect is used as the active materials for improving the thermoelectric response but reducing the device footprint down to sub-wavelength scale in one direction, which shows great promise for compact devices.

2: The performance characterization of our photodetectors is new. Most of previous works just use polarization ratio to evaluate the performance of polarization-sensitive photodetectors which is not accurate enough, whereas our work proposed a combined method to evaluate simultaneously two important figures of merit (FoM) i.e. the polarization sensitivity and detectivity. As a result, our method is more accurate than all previous approaches, making the polarization-sensitive photodetector we designed highly suitable for a number of practical applications, e.g. 1) when applied to the polarization angle sensitivity (PAS), our photodetector can push the sensitivity limit to 7.1 V/W•degree; 2) when applied to polarization angle detectivity (PAD) our photodetector has a sensitivity of 2.9×10^5 Jones/degree. As we demonstrate experimentally, these two FoM (as proposed in our work) are more general than the polarization ratio (widely used in previous works), because the polarization ratio loses some important information such as responsivity and detectivity, especially for performance evaluation and comparison between devices with different materials and different configurations. With a consideration of both polarization ratio and responsivity, the proposed photodetectors in our work show an ultrahigh polarization sensitivity, which is one order of magnitude higher than those reported in the literature.

Comment 1: Polarization detector (polarimeter) and polarization detector arrays have been demonstrated based on different device configurations. The authors failed to mention in their introduction some compact solutions reported in literature, some of which in IR and LWIR wavelengths are already commercially available (<https://www.polarissensor.com/pyxis/>). This imaging sensor makes use of chip integrated plasmonic nanogratings on the LWIR detector array to achieve high linear polarization selection to measure S0, S1, S2 at each imaging pixel. In this manuscript, the authors make use of Plasmonic metasurface absorbers to introduce polarization dependent detector results, which is essentially the same concept. Detection of linear polarization with very high selectivity is not a challenging topic anymore. Now in research field, major interests are focus on achieve the detection of circular polarization components, i.e., S3. Many metasurface solutions have been provided to achieve complete measurements of full-stokes

polarization detection. The following are just some examples, not all.

Pors, Anders, Michael G. Nielsen, and Sergey I. Bozhevolnyi. "Plasmonic metagratings for simultaneous determination of Stokes parameters." *Optica* 2.8 (2015): 716-723.

Li, W.; Coppens, Z. J.; Besteiro, L. V.; Wang, W.; Govorov, A. O.; Valentine, J. Circularly Polarized Light Detection with Hot Electrons in Chiral Plasmonic Metamaterials. *Nat. Commun.* 2015, 6, 8379

Bai, J.; Wang, C.; Chen, X.; Basiri, A.; Wang, C.; Yao, Y. Chip-Integrated Plasmonic Flat Optics for Mid-Infrared Full-Stokes Polarization Detection. *Photonics Res.* 2019, 7 (9), 1051– 1060,

Afshinmanesh, F.; White, J. S.; Cai, W.; Brongersma, M. L. Measurement of the Polarization State of Light Using an Integrated Plasmonic Polarimeter. *Nanophotonics* 2012,1 (2), 125,

Response to reviewer:

We are thankful to the reviewer for the kind recommendation. We agree with the reviewer that substantial efforts in research field have been focused on achieving the detection of circular polarization components, i.e., S_3 , including our work [*Nat. Commun.* 13, 4560 (2022)] and a few others [*Nature Photonics* 17, 171–178 (2023); *ACS Nano* 2020, 14, 12, 16634–16642]. As we mentioned in the reply to earlier questions and as discussed in the manuscript, it's still a challenge to realize photon detection with an ultrahigh polarization sensitivity in the long-wave infrared range (as shown in Fig. 3e and Supplementary Table 2). Even with the commercial devices mentioned above mentioned above by the referee, it is still difficult to achieve a polarization sensitivity $> 1 \text{ V/W} \cdot \text{degree}$. In this work, we realized a high polarization sensitivity ($7.10 \text{ V/W} \cdot \text{degree}$). Our designed IR detector comprises a perfect plasmonic absorber with the finite-size effect and a 1D Te nanoribbon with a large Seebeck coefficient. Particularly, a peak polarization angle sensitivity of $7.10 \text{ V/W} \cdot \text{degree}$, which is one order of magnitude higher than those reported in the literature, is achieved in the long-wave infrared region ($\lambda=8.0 \mu\text{m}$). On the other hand, the metasurface is integrated with the active thermoelectric materials in our detector configuration, and this can reduce the crosstalk between adjacent pixels (*Opto-Electronic Advances* 2022, 5 (11), 220058-220058), which is a fundamentally different concept to the imaging sensor (<https://www.polarissensor.com/pyxis/>) even though the configurations show some similarity relevance.

In addition, for research interests in this community, we believe that both the linear and circular polarization components are at research frontlines. The following are just some

recent examples on linear polarization sensitive photodetectors, not exclusive.

(1) Wei, J.; Xu, C.; Dong, B.; Qiu, C.-W.; Lee, C., Mid-infrared semimetal polarization detectors with configurable polarity transition. *Nat. Photonics* 2021, 15 (8), 614-621.

(2) Jiao, H.; Wang, X.; Chen, Y.; Guo, S.; Wu, S.; et. al., HgCdTe/black phosphorus van der Waals heterojunction for high-performance polarization-sensitive mid-wave infrared photodetector. *Science Advances* 2022, 8 (19), eabn1811.

(3) Wu, S.; Chen, Y.; Wang, X.; Jiao, H.; Zhao, Q.; et. al., Ultra-sensitive polarization-resolved black phosphorus homojunction photodetector defined by ferroelectric domains. *Nat. Commun.* 2022, 13 (1), 3198.

Comment 2: Materials with claimed large Seebeck coefficients have been studied extensively in literature, such as black phosphor. Some of the materials were reported with comparable or even higher values than the claimed value (>400 $\mu\text{V/K}$) for Te nanoribbons. The authors did not explain clearly and prove why they choose Te nanoribbons? What are the criteria for material selection and how the study proved the hypothesis?

Response to reviewer: We thank the reviewer for pointing out the aspects which helps us to improve our manuscript. In principle, the active materials should also possess a low thermal conductivity and a high carrier mobility, in addition to a large Seebeck coefficient. In this work, we choose 1D Te nanoribbon as the active material owing to its advantages in thermoelectric material, such as ultralow thermal conductivity (2.16 $\text{W/m}\cdot\text{K}$), a high Seebeck coefficient (413 $\mu\text{V/K}$), a high carrier mobility (993 $\text{cm}^2/\text{V}\cdot\text{s}$), and a good electrical conductivity owing to its narrow bandgap. In addition, comparing with 2D nanosheets, the 1D Te NR based device can realize a higher photoresponse and a smaller device footprint, simultaneously. We thank the referee for mentioning the black phosphor. Although it has a comparable Seebeck coefficient and a higher carrier mobility, its thermal conductivity (15 $\text{W/m}\cdot\text{K}$) is relatively high. Besides, its poor thermos- and environmental stabilities greatly limit its applications to the photothermoelectric detectors [*Nano letters* 2016, 16, 4819; *Adv. Mater.* 2018, 1704749]. On the other hand, we have also chosen three others different 2D materials as the active materials in our proposed device configuration and compare their performance as shown in Supplementary Fig. 17. According to reviewer's suggestion, we added a clear corresponding description in the Revised Manuscript, also copied below:

On page 7 of the revised Manuscript.

Apart from the large optical anisotropy used for supporting a large PR, the thermoelectric material with large Seebeck effect, low thermal conductivity, and high carrier mobility, is also desirable to achieve a high responsivity. Benefiting from the high tolerance for selection of thermoelectric materials for the as-proposed PTE mechanism, we select Te nanoribbon as the active material owing to its advantages, such as ultralow thermal conductivity (2.16 W/m•K) due to its heavy atom mass²¹, a high Seebeck coefficient (413 $\mu\text{V}/\text{K}$) boosted by the quantum confinement effect induced sharp shapes of the density of states at band edges^{22, 23}, and a good electrical conductivity owing to its narrow-bandgap²⁴.....

On the page 12 of the revised Manuscript.

.....On the other hand, other different 2D materials (PdSe₂, MoS₂, InSe) were used as the active materials for the proposed detector design. The responsivities are 27, 62, and 26 V/W for PdSe₂, MoS₂, and InSe based devices with the same device configurations, respectively (Supplementary Fig.17). Our previous work also demonstrated that the 2D PdSe₂ nanosheet based detector shows a better performance than that of BP and Graphene based devices⁷. These results further indicate that the 1D Te NR is more suitable for the photothermoelectric detection with our proposed device architecture.

On page 18 of the revised Supplementary Information.

Supplementary Figure 17: Photoresponses of designed devices with different active 2D materials. a, Optical image and corresponding polarization-sensitive photoresponses of

PdSe₂ based detector. **b**, Optical image and corresponding polarization-sensitive photoresponses of MoS₂ based detector. **c**, Optical image and corresponding polarization-sensitive photoresponses of InSe based detector. The light blue area indicates the optical cross-section area of the devices. Scale bar: 10 μm. Blue dots are measured data, and red lines are fitting curves.

Comment 3: The authors claimed a linear polarization extinction ratio of infinity, which is highly questionable. Usually, one obtains the linear polarization extinction ratio by taking the ratio between the detector output voltage/current for the favored linear polarization and that of its orthogonal polarization. A ratio of infinity means one reads exactly 0 for the orthogonal polarization in experiments, which does not seem to be reasonable if the measurement instrument has sufficient level of accuracy.

Response to reviewer: We thank the reviewer for the question. We agree that it is hard to read exactly zero photoresponse if the measurement instrument has sufficient level of accuracy. In our experiments, the photovoltage responses for $\theta=0^\circ$ are unable to distinguish from the noise owing to the limited level of the equipment accuracy (Keysight, B2912A; voltage resolution: 100 nV). The polarization ratio is estimated to be in the range of 10^4 - 10^5 when the incident light power is in the range of 2.76-25.7 μW, which is far higher than the most previous reported polarization ratio. Therefore, to be more rigorous, we use “near-infinite polarization ratio ($PR \rightarrow \infty$)” in the Revised Manuscript to replace the “infinite polarization ratio ($PR = \infty$)” in the original version.

Comment 4: The D^* of the device is not very good compared with the commercially available LWIR detectors. The detection speed is not very impressive, either. I understand that this is just a prototype device. However, the authors did not provide analysis of the ultimate limits of device performance. Is it fundamentally possible to achieve better performance than the state of art LWIR detectors? If so, what does it take to get there?

Response to reviewer: We thank the reviewer for the constructive suggestion. According to the reviewer’s suggestion, we added corresponding discussions about the limitations and potentials of our device performance, providing some potential strategies to further improve the device performance. The corresponding discussions have been added in the revised manuscript, also copied below:

On page 14 of the revised Manuscript.

.....However, comparing with commercially available IR detectors, the detectivity and response speed of our detector still need to be further improved (Supplementary Table 2).

On page 20 of the revised Manuscript.

.....Last but not the least, the detector performance in this work can be further improved, especially in terms of the response speed and the detectivity. Based on the photothermoelectric response mechanism, the performance of the detector can be further improved by systematically optimizing the electrode metal with lower contact barrier, the doping density of thermoelectric materials with higher Seebeck coefficient, the plasmonic metal microstructure with lower heat capacity and higher photothermal effect, the channel length matched better with the thermal decay length, and so on.....

On page 23 of the revised Supplementary Information.

Supplementary Table 2. Performance comparison of linear polarization sensitive photodetectors operating in mid-/long-wave infrared region

Materials	Mechanism	Wavelength (μm)	PR	Response time (s)	Responsivity (V/W)	Detectivity (Jones)	PAS (V/W*degree)	PAD (Jones/degree)	Ref.
InAsSb	PVE	3-10		3x10 ⁻³	1.9 mA/W	8.5x10 ⁷			2
MCT	PVE	2-10		10x10 ⁻⁶	185	2x10 ⁸			3
Gr/Au antenna	PTE	6.6	1.5	17x10 ⁻⁶	92	/	0.53	/	34
Gr/Au antenna	PTE	4.0	∞	<667x10 ⁻⁶	15.6	1.56x10 ⁶	0.27	2.7x10 ⁴	8
Gr/Au antenna	BPVE	4.0	-1		27	5x10 ⁶	0.94	1.7x10 ⁵	31
Twist Gr	BPVE	7.7	-2	<100x10 ⁻⁶	3.7	/	0.09	/	33
PdSe ₂ /Au metamaterials	PTE	5.3	∞	76x10 ⁻⁶	3.6	2.5x10 ⁵	0.06	4.1x10 ³	7
PdSe ₂	PTE	4.6	2.03	52x10 ⁻⁶	21.6	6.7x10 ⁶	0.19	5.9x10 ⁴	16
		10.5	1.21		14.3	4.5x10 ⁶	0.04	1.2x10 ⁴	
TaIrTe ₄	Shift Current	4.0	8.5		53.3	/	0.82	/	32

Te/Au metamaterials	PTE	8.0	∞	176×10 ⁶	410	1.7×10 ⁷	7.1	2.9×10 ⁵	This work
			-1		380	1.1×10 ⁷	13.2	3.8×10 ⁵	

PVE: photovoltaic effect. PTE: photothermoelectric, BPVE: bulk photovoltaic effect.

PAS: polarization angle sensitivity. PAD: polarization angle detectivity.

a: P13894-011MA from HAMAMASTU. b: PDAVJ10 from THORLABS.

Ref. is cited in the main article.

Comment 5: When calculating of NEP and D^* , the authors estimated the power incident onto their devices using the ratio between the device area and the optical spot size. This method is a bit questionable because the device optical cross-section is usually larger than the device area. Besides, it is necessary for the authors to provide all the parameter values used for estimating NEP and D^* , including the device area A.

Response to reviewer: We thank the reviewer for pointing out this issue which helps us to improve our manuscript. To estimate the device performance reasonably, two kinds of device areas including the optical cross-section area and the electrical area of the device should be used (Nature 556, 85–88 (2018)). In the revised manuscript, we use the device optical cross-section area (S) to calculate the incident power for responsivity (R), and use the detector area (S_{det}) to calculate the specific detectivity. According to the reviewer's suggestion, we provide all the parameters values used for calculating the S , S_{dec} , R , NEP, and D^* . To describe this issue clearly, we added corresponding description in the revised Manuscript, also copied below:

On page 11 of the revised Manuscript.

.....where, P_0 is the total power of the incident light, r_1 and r_2 are two axis radii of the Gaussian beam, and S is the device optical cross-section area as shown in Fig. 3a. Here, the device optical cross-section area for ports 1-3 is 68, 102, and 136 μm^2 , respectively, which is calculated by: $S=L \times P$, where L is the channel length, P is the period.....

On page 13 of the revised Manuscript.

.....Moreover, the detector also exhibits a low dark noise spectral density (S_n) down to 17 nV $\text{Hz}^{-1/2}$ at a high frequency range (over 5 kHz) corresponding to a noise-equivalent power of

0.04 nW Hz^{-1/2} (Supplementary Fig. 14). The specific detectivity of the detector is calculated to be 1.7×10⁷ Jones at room-temperature by $D^* = R\sqrt{S_{\text{det}}}/S_n$, where S_{det} is the area of the detector, R is the responsivity, and S_n is the noise spectral density. Here, the detector area S_{det} for Port 1 is 49 μm².....

REVIEWER COMMENTS

Reviewer #1 (Remarks to the Author):

The revised manuscript has improved by addressing the technical questions well.

Reviewer #2 (Remarks to the Author):

The authors have addressed most of the issues and explained the advantages of using 1D tellurium nanoribbon for mid-IR detection. Given the results they presented, it could be further explored to enable room temperature mid-IR photodetectors with reasonably good detectivity and more importantly higher speed than some state-of-the-art mid-IR photodetectors.

I understand that polarization imaging is important and chip-integrated full stokes polarization imaging sensors would be very attractive. But the linear polarization detection is indeed quite trivial at this point.

Furthermore, the claim of a near-infinite polarization ratio ($PR \rightarrow \infty$) does not seem to be scientifically true. Theoretically it is impossible to achieve infinite polarization extinction ratio using the proposed nanostructures even based on full wave simulation results.

If the authors do want to make a claim about their high LP extinction ratio in experiment, they shall use the right equipment with sufficient sensitivity and if necessary try to increase the input light intensity to be able to obtain meaningful results instead of only changing the words to near-infinity.

What does near-infinity mean? Does it mean larger than 100, 1000, 10000, or even higher? Some LP polarizers in the market were able to demonstrate polarization extinction ratio over 10K.

Reply to the Reviewers' Comments

We are grateful again to the constructive comments and advice provided by the reviewers on our manuscript Nature Communications (ID: NCOMMS-22-43717A. Title: Long-wave infrared photothermoelectric detectors with ultrahigh polarization sensitivity). In response to these comments, we have provided necessary discussions and analyses, and revised the manuscript and the supplementary information thoroughly. We believe that we have addressed all the comments raised by the reviewers.

The corresponding amendments addressing the referees' concerns have been marked with yellow background for easy reference in the revised manuscript (Revised Manuscript with marks) and revised Supplementary Information (Revised Supplementary Information with marks). Our replies to the comments by the reviewers and the corresponding changes are listed below.

The following are details of our point-by-point responses to reviewers' comments:

Reviewer #1:

Comments:

The revised manuscript has improved by addressing the technical questions well.

Response to reviewer: We are grateful to the reviewer for the review and appreciate the reviewer's positive comments on our work.

Reviewer #2:

Comments:

The authors have addressed most of the issues and explained the advantages of using 1D tellurium nanoribbon for mid-IR detection. Given the results they presented, it could be further explored to enable room temperature mid-IR photodetectors with reasonably good detectivity and more importantly higher speed than some state-of-the-art mid-IR photodetectors.

Response to reviewer: We are grateful to the reviewer for the review and appreciate the reviewer's constructive suggestions that helped us to improve the quality and depth of our

work.

Comment 1: I understand that polarization imaging is important and chip-integrated full stokes polarization imaging sensors would be very attractive. But the linear polarization detection is indeed quite trivial at this point.

Response to reviewer:

We are thankful to the reviewer for the kind recommendation. For the polarization imaging, both the angle of linear polarization (AoLP) and degree of linear polarization (DoLP) are two usually used parameters to obtain additional information relating to material properties and illumination conditions, which are based on linear polarization detection (*Opt. Eng.* 58, 082419 (2019); *Opt. Lett.* 40, 882–885 (2015)). On the other hand, high polarization sensitivity (include large PR and high responsivity) is highly critical for improving the accuracy of detection in practical applications. However, most previously reported linear polarization-sensitive photodetectors based on natural or artificial materials usually exhibit unipolar polarization-dependent photoresponses and the corresponding PR is generally small, e.g., $0 < PR < 100$ (*Nat. Nanotech.* 10, 707-713 (2015); *Nat. Photonics* 12, 601-607 (2018); *Light: Sci. Appl.* 7, 20 (2018)). In this work, even though we focus on linear polarization detection, we realize an ultrahigh polarization ratio ($PR=2.5\times 10^4$) and a high photoresponsivity (410 V/W), thereby achieving a high polarization sensitivity (7.1 V/W•degree) for the long-wave infrared regime. Therefore, for the point of polarization imaging, we believe that our work provides a reliable strategy for developing polarization-resolved photodetectors for the mid-infrared polarization imaging.

Comment 2: Furthermore, the claim of a near-infinite polarization ratio ($PR\rightarrow\infty$) does not seem to be scientifically true. Theoretically it is impossible to achieve infinite polarization extinction ratio using the proposed nanostructures even based on full wave simulation results.

If the authors do want to make a claim about their high LP extinction ratio in experiment, they shall use the right equipment with sufficient sensitivity and if necessary try to increase the input light intensity to be able to obtain meaningful results instead of only changing the words to near-infinity.

What does near-infinity mean? Does it mean larger than 100, 1000, 10000, or even higher? Some LP polarizers in the market were able to demonstrate polarization extinction ratio

over 10K.

Response to reviewer: We thank the reviewer for the kind suggestion. In our experiments, the photovoltage responses were recorded by the source measurement unit with a high-level equipment accuracy of 100 nV (Keysight, B2912A). On the other hand, based on the spectral voltage noise density measured by a lock-in amplifier (Zurich Instruments, HF2LI) with a typical input voltage noise of 5 nV/Hz^{1/2}, the voltage noise level of our device at low frequency (100 Hz) is about 410 nV/Hz^{1/2} (as shown in Supplementary Figure 14). As a result, the ultralow photovoltage responses at $\theta=0^\circ$ are unable to be distinguished from the device noise. Based on the device voltage noise at a low frequency of 100 Hz, the maximal polarization ratio is calculated to be about 2.5×10^4 when the incident light power at device is 25.7 μW (corresponding to a light power density of 37.6 W/cm²), which is much higher than most previously reported polarization ratios. According to reviewer's suggestion, we replaced the "a near-infinite PR" by "an ultrahigh PR (2.5×10^4)" in the revised Manuscript and added a clear corresponding description in the Revised Manuscript, also copied below:

On page 13 of the revised Manuscript.

.....The maximal polarization ratio is about 2.5×10^4 when the incident light power at device is 25.7 μW , which is calculated based on the device voltage noise at a low frequency of 100 Hz (Supplementary Fig. 14).....